# Nowcasting the 2022 mpox outbreak in England

**Christopher E. Overton**[1,2,3]*, **Sam Abbott**[4], **Rachel Christie**[2], **Fergus Cumming**[2], **Julie Day**[2], **Owen Jones**[2], **Rob Paton**[2], **Charlie Turner**[5], **Thomas Ward**[2]

**1** Department of Mathematical Sciences, University of Liverpool, Liverpool, United Kingdom, **2** UK Health Security Agency, Data Science and Analytics, London, United Kingdom, **3** Department of Mathematics, University of Manchester, Manchester, United Kingdom, **4** London School of Hygiene and Tropical Medicine, London, United Kingdom, **5** UK Health Security Agency, Mpox Data, Epi and Analytics Cell, London, United Kingdom

* c.overton@liverpool.ac.uk

**Data Availability Statement:** Training data for the modified data used to inform the best fitting models is available at https://github.com/OvertonC/nowcasting_the_2022_mpox_outbreak_in_england. This data has had dates removed to

## Abstract

In May 2022, a cluster of mpox cases were detected in the UK that could not be traced to recent travel history from an endemic region. Over the coming months, the outbreak grew, with over 3000 total cases reported in the UK, and similar outbreaks occurring worldwide. These outbreaks appeared linked to sexual contact networks between gay, bisexual and other men who have sex with men. Following the COVID-19 pandemic, local health systems were strained, and therefore effective surveillance for mpox was essential for managing public health policy. However, the mpox outbreak in the UK was characterised by substantial delays in the reporting of the symptom onset date and specimen collection date for confirmed positive cases. These delays led to substantial backfilling in the epidemic curve, making it challenging to interpret the epidemic trajectory in real-time. Many nowcasting models exist to tackle this challenge in epidemiological data, but these lacked sufficient flexibility. We have developed a nowcasting model using generalised additive models that makes novel use of individual-level patient data to correct the mpox epidemic curve in England. The aim of this model is to correct for backfilling in the epidemic curve and provide real-time characteristics of the state of the epidemic, including the real-time growth rate. This model benefited from close collaboration with individuals involved in collecting and processing the data, enabling temporal changes in the reporting structure to be built into the model, which improved the robustness of the nowcasts generated. The resulting model accurately captured the true shape of the epidemic curve in real time.

## Author summary

During 2022, outbreaks of mpox, the disease caused by the monkeypox virus, occurred simultaneously in multiple non-endemic countries, including England. These outbreaks were distinct from historic outbreaks with a majority of cases in gay, bisexual and other men who have sex with men and in individuals without recent travel histories to endemic countries. To inform public health policy, understanding the number of new cases and

improve anonymity. This data enables the non-parametric models to be used, and can also be used as training data for future development of nowcasting models. Code for running all models is available at https://github.com/OvertonC/nowcasting_the_2022_mpox_outbreak_in_england. Individual-level data on the reporting delay used to inform the parametric models has not been made available due to potential identifiability. An application for data access can be made to the UK Health Security Agency. Data requests can be made to the Office for Data Release (https://www.gov.uk/government/publications/accessing-ukhsa-protected-data/accessing-ukhsa-protected-data) and by contacting DataAccess@ukhsa.gov.uk. All requests to access data are reviewed by the Office for Data Release and are subject to strict confidentiality provisions in line with the requirements of: the common law duty of confidentiality, data protection legislation (including the General Data Protection Regulation), Caldicott principles, the Information Commissioner's statutory data sharing code of practice, and the national data opt-out programme.

**Funding:** This work was supported by the Wellcome Trust (210758/Z/18/Z to SA). The funders had no role in study design, data collection and analysis, decision to publish, or preparation of the manuscript.

**Competing interests:** The authors have declared that no competing interests exist.

growth rate of the outbreak in real-time is essential. However, the outbreak was characterised by long delays from individuals developing symptoms (or getting a test) and being reported as a positive case. This creates a biased picture of the outbreak, where observed real-time cases underestimates the true extent of the outbreak. We developed a mathematical model that accounts for these reporting delays to estimate the true shape of the epidemic curve in real-time. The modelled outputs are able to accurately capture the true shape of the epidemic, and provide improved real-time insight over the raw data. This model was used continuously throughout the outbreak response in the UK to provide insight to the incident management team at the UK Health Security Agency.

## 1. Introduction

Mpox, a disease caused by the monkeypox virus [1], was first discovered in 1958 when researching monkeys that showed signs of a poxvirus [2]. In 1970, the first documented human cases of mpox were detected in the Democratic Republic of the Congo [3]. Since then, the disease has become endemic in the DRC and spread to other Central and West African countries. Despite being endemic, outbreaks of mpox have often been zoonotic in origin [4]. Prior to 2022, cases and transmission of mpox have occurred outside of Central and West African countries, but these have usually consisted of a primary case with recent travel history to endemic regions, followed by self-limiting local outbreaks. In May 2022, cases of mpox were detected in multiple non-endemic countries.

The detection of cases in May 2022 quickly demonstrated signs atypical of the usual mpox outbreaks, since multiple cases were detected in many countries simultaneously, and most cases had no known travel links to endemic regions. The majority of cases, within this outbreak, have occurred in gay, bisexual and other men who have sex with men (GBMSM). Case questionnaire data suggests likely close sexual contact as a driver of transmission, due to the prolonged skin-to-skin contact. The international dispersion of the virus has resulted in the largest outbreak of mpox reported outside of Africa and in July 2022 the WHO declared the outbreak of mpox a Public Health Emergency of International Concern (PHEIC)[5].

As part of the surveillance aspect of the public health response in the UK, it is vital to understand the shape of the epidemic curve, to ensure public health teams are aware of the current epidemic trajectory and case burden, and to facilitate evaluation of interventions in real time. Ideally, when monitoring an epidemic curve, one would look at the infection date of individuals. However, infection date is rarely observed directly. Instead, other variables need to be used as a proxy for infection date, each of which are lagged relative to infections. The shorter the lag, the more useful the proxy since the shape of the epidemic curve will be less perturbed and the epidemic curve will be more closely aligned temporally. However, events that occur closer to the time of infection may be more affected by reporting delays, in that the delay from the event occurring to being reported to public health teams may be longer. This can bias the epidemic curve when using these proxy events in real-time. Nowcasting is an area of research that attempts to correct for this reporting delay bias when reproducing the epidemic curve.

There are a range of nowcasting tools and packages available [6–11]. However, in any nowcasting problem, the unique characteristics of the local setting and data processing mean that a general tool may not be suitable. For example, through working directly with data processing teams at UKHSA, we were able to understand the data reporting cycle and temporal changes, which is important to build directly into the model. Also, having access to individual-level data enabled direct estimation of the reporting delay distributions, so it was useful to have flexible

models where these distributions could be incorporated directly into the model. In addition, computational tractability and reliability were vital, to ensure modelling products could be delivered every week. Another challenge resulted from many of the existing packages requiring Bayesian implementation, which could not be run on the same IT systems used to access the data, hindering the development of efficient modelling pipelines, which is essential for operational product delivery. Therefore, we developed a nowcasting model that makes novel use of individual-level patient data using hierarchical generalised additive models (GAM), based on the Chain Ladder nowcasting method [12]. These models have a similar structure to GAM-based nowcasting models used in the German COVID-19 nowcasting hub [13], but we consider both parametric and non-parametric approaches to the reporting delay distribution, making optimal use of the available individual-level patient data. These are easy to implement and run, but are not as theoretically robust as the Bayesian approaches. However, the developed model is easy to use and has sufficient flexibility to modify in real-time the evolving data landscape. This made the model robust to changes in data processing, which allowed continuous delivery of the operational nowcasting product throughout the mpox outbreak, with the outputs regularly being included in the UKHSA monkeypox technical briefing [14].

A further challenge in real-time nowcasting is when the reporting structure changes. Nowcasting fits a relationship to the historic reporting structure and assumes this remains the same going forwards. Time-varying reporting structures can be fitted, but this requires sufficient data to pick up recent changes in the trend. Therefore, at times when the reporting structure changes, the nowcasting can be compromised. To get around this, we perform data modification, to change the historic data to reflect the current reporting structure. This involves discussing details around the data reporting changes with those involved in data collection and processing, to understand how the reporting structure is changing for the recent data. This close collaboration with those collecting/curating the data is vital for building nowcasting models in real time as part of an operational outbreak response, and we show that not taking this knowledge into account reduces the performance of our models.

Another challenge arises when ascertainment rate is changing over time. For example, ascertainment rate of symptom onset dates is likely to be high at the start of an outbreak, when public health teams can dedicate a lot of time to each individual case. However, as case numbers rise, it may not be feasible to continue such an intense individual-level response. This can lead to the ascertainment rate of data on symptom onset declining as case numbers rise, which causes artificial damping of the epidemic curve. To adjust for this, one would need methods that simultaneously adjust for nowcasting and changing case ascertainment rates. This is an important area for future research into nowcasting, but we do not account for it here due to the time pressures involved in getting an operational product.

In this paper, we develop a novel nowcasting model using generalised additive models, implemented in the mgcv [15] package in R [16]. We propose a range of models, ranging from fully non-parametric to hybrid models using a mixture of parametric and non-parametric aspects, which we compare using coverage, bias, and weighted interval scores using the scoringutils [17] package. We then discuss a range of challenges that have emerged in real-time during the nowcasting of the mpox outbreak, and describe the solutions we have applied.

## 2. Data

### 2.1 Data sources

Data on daily cases are obtained from the mpox case linelist. The mpox case linelist is compiled Monday to Friday by Outbreak Surveillance Team and South East and London Field Service team using testing data from MOLIS (Molecular Laboratory Information System) and SGSS

(Second Generation Surveillance System), combined with operational data from HPZone collected by Health Protection Teams. The line list provides information about cases such as symptom onset date (the date of the first symptom, from HPZone), specimen date (the date the specimen was taken for the test, from MOLIS/SGSS) and the date confirmed cases were reported to the mpox Incident Management Team (IMT).

The mpox linelist comprised of 3,817 cases as of 31 December 2022. In this paper, we focus on the acute phase of the outbreak, running the modelling until 31 August 2022, where 3,484 cases had been reported. These data are biased towards males and age groups 25–34 and 35–44 (Table 1). We remove individuals who report recent foreign travel (964 as of 31 December 2022), since many of these individuals may have been infected abroad.

## 2.2 Data coverage

In the nowcasting, we consider two key variables by which the epidemic curve will be visualised. These are specimen date, the date a positive swab is taken, and symptom onset date, the date a positive individual develops symptoms. Specimen date has a high completion rate, being above 90% for most of the outbreak (Fig 1(A)). At the start of the outbreak, symptom onset date also had a high completion rate. However, this relied on labour intensive questionnaires from local health protection teams. As the outbreak grew in size, the policy changed and questionnaires became optional. This led to a sharp drop in coverage, from over 90% before June 2022 to around 50% from mid July 2022 (Fig 1(B)). Over June 2022, the coverage was continually declining, which will bias the shape of the observed epidemic curve by symptom onset date during June.

## 2.3 Data limitations

In addition to the data coverage issues (Section 2.2), there are other limitations to the data which create challenges for the nowcasting. The first issue concerns nowcasting by symptom onset date. When nowcasting an epidemic curve, it is essential to have reliable estimates of the delay from the event of interest occurring to this event being reported in the data. However, sometimes individual cases may be reported before data are available on all events of interest, with these data being added to the record over time. This means that the "date reported" variable for an individual

**Table 1. Demographics of confirmed mpox cases from the mpox linelist (as of 31 August 2022 and 31 December 2022).** For statistical disclosure, counts under 10 are removed.

| Demographics | Number of cases | |
| --- | --- | --- |
| | 31/08/2022 | 31/12/2022 |
| **Gender** | | |
| Male | 3,388 | 3,700 |
| Female | 48 | 63 |
| Undefined | 48 | 54 |
| **Age group** | | |
| 0–15 | - | - |
| 16–24 | 225 | 254 |
| 25–34 | 1231 | 1338 |
| 35–44 | 1188 | 1303 |
| 45–54 | 564 | 618 |
| 55–64 | 215 | 231 |
| 65–74 | 37 | 46 |
| 75 + | - | - |
| NA's | 21 | 22 |

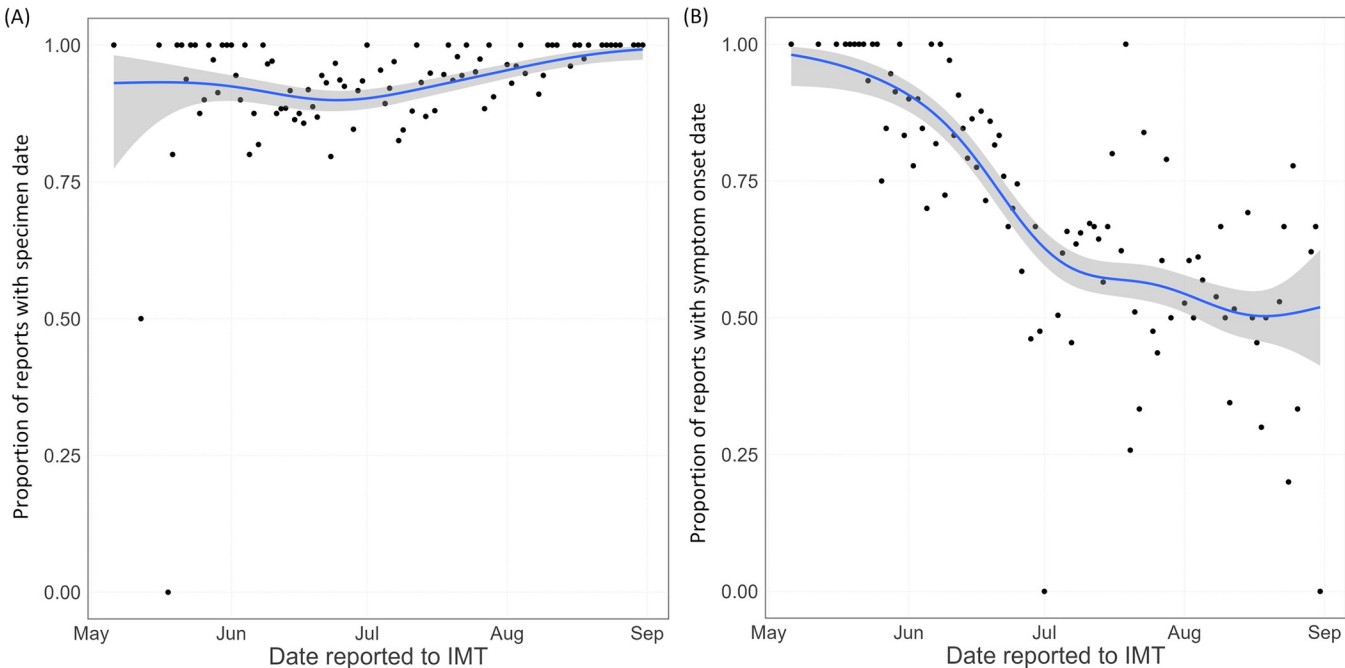

**Fig 1. Coverage of the key variables used in the nowcasting during the study period.** Black points show the data, the blue line is a binomial GAM smooth through the data. (A) shows the proportion of reported cases with specimen date recorded, plotted by reporting date. (B) shows the proportion of reported cases with symptom onset date recorded, plotted by reporting date.

may not correspond to the "date event reported" for that individual. This presents a challenge for nowcasting, since it means that the historic delays from "event" to "date reported" may not reflect the actual distribution of delays from "event" to "date event reported", which governs how the epidemic curve evolves in real time. One solution to this is to have daily timestamps of the database, which track the state of data in real time. However, often these may not be available due to database design constraints, which is the case for the UK mpox data.

The second issue concerns nowcasting by specimen date. As the epidemic progressed, different testing laboratories have been opened, and test processing procedure has changed. The main change that has been introduced relates to increasing the processing requirements before confirming a test. As of 30 June 2022, test processing procedure was changed to add an extra day of processing before they enter the UKHSA database. In addition to this, as the outbreak progressed, processing over the weekend was changed, to allow for reduced staffing levels over the weekend. This led to tests no longer being processed on a Sunday, which changed the weekly reporting cycles. This mainly affects specimens collected on Fridays and Saturdays, which will have approximately 1 extra day delay relative to earlier samples. Specimens collected on other days may have also been affected, but in general the reporting labs intended to process specimens within the two days after the specimen was collected, so Friday and Saturday are most sensitive to this change.

## 3. Methods

### 3.1. What is nowcasting?

Nowcasting aims to estimate the number of events occurring on a given day that are yet to be reported [18]. Reported data can often be considered as a "reporting triangle". That is, for data sufficiently long ago, we have subsequent reports starting on that day and running up to today.

For each subsequent day, we have one fewer day of reporting, up until the most recent day, where we only have data reported on that day. Therefore, the reported data forms a triangle. Nowcasting aims to complete this reporting triangle by turning it into a reporting square, where all observation days have the full range of subsequently reported values.

### 3.2. Nowcasting model

The number of events occurring on a given day, $y(o)$, is the sum over all possible reporting delays of the number of events that occurred on day $o$ and were reported after $d$ days,

$$y(o) = \sum_{d=0}^{\infty} y_d(o).$$

The summand here is equal to the sum over all events that occurred on day $o$ of the probability that the event is reported after $d$ days, i.e.

$$y_d(o) = \sum_{1}^{y(o)} f_\theta(d) = y(o) \times f_\theta(d),$$

where $f_\theta(d)$ is the probability that an event is reported after $d$ days. Since both the epidemic and reporting process are stochastic, we assume there is some error in the reported data. This error is proportional to the magnitude of the observations, since the stochasticity acts on an individual basis, so we assume the observations are given by $y_d(o) \times \epsilon$. Since we are modelling an epidemic, we assume that the number of events each day follows an exponential function in time. However, instead of assuming a fixed exponential rate, we allow the exponential rate to be a smooth function of time. That is, the number of events on day $o$ is given by

$$y(o) = y_0 \exp(s(o)),$$

for a smooth function $s(\cdot)$. For the probability of a reporting delay of length $d$, $f_\theta(d)$, we can consider a few approaches:

1. The probability can be fit independently for each delay length, for example using a generalised linear model.

2. The probability can be assumed to follow a smooth function of the delay lengths, which can be fit using a spline.

3. A parametric model for the reporting delay can be assumed.

In this paper, we focus on approaches (2) and (3), since it is reasonable to assume that the reporting delay distribution is a smooth function of time. Approach (1) is the approach used in the Mack Chain Ladder method [12].

To allow the reporting delay distributions to change over time, we consider a maximum value, $T$, for the reporting delay. Above this value, we assume the probability of an event being reported to be equal to zero. This means that the model is fitted using data from the last $T$ days, so will reflect the reporting delay over this time period, rather than the whole outbreak. In addition to capturing time varying reporting trends, this assumption also reduces the computational time of the model, which makes it more efficient for operational use. Therefore, we have

$$y(o) = \sum_{d=0}^{T-1} y_d(o).$$

When selecting a suitable value for $T$, we need to ensure that it is sufficiently large to capture all likely reporting delays. Additionally, to aid computational time, we only fit data using the last $T$ days, rather than only truncating the delay distribution at $T$. Therefore, we also want $T$ to be large enough for the model to identify trends in the data.

### 3.3. Smooth non-parametric model

Under the assumption that the reporting delay can be described by a smooth function of time, we assume that $f_\theta(d) \approx \exp(s(d))$, which leads to

$$y_d(o) = y_0 \exp(s(o)) \exp(s(d)) \exp(\epsilon_{od}).$$

To solve this, we can use a generalised additive model with a negative binomial error structure,

$$\log(y_d(o)) \sim \beta_0 + \mathrm{wday}(o) + \mathrm{wday}(d) + s_1(o) + s_2(d) + \epsilon_{od},$$

where $s_1(\cdot)$ and $s_2(\cdot)$ are smoothers. Here, $\mathrm{wday}(o)$ and $\mathrm{wday}(d)$ are random effects to capture the day of week cycle in the epidemic curve and reporting trends, respectively. Negative binomial error is chosen because the daily case counts are integer valued and overdispersed (since they are generated from a stochastic process). For the smoothers, we need to specify the type and number of basis functions to be used. We use penalised cubic regression splines for the type, and the number used depends on the length of the data series. For the epidemic curve (event date smoother), we add one basis function for every $\tau$ dates, which effectively allows the epidemic to change shape every $\tau$ days. We consider two different values of $\tau$, $\tau \in \{7, 14\}$, and evaluate the performance to select the optimal number of basis functions. When selecting for $\tau$, the model is trading off between flexibility (smaller $\tau$) and reducing sensitivity to the part of the data with substantial backfilling (larger $\tau$). Since the last part of the data is highly sensitive to the backfilling, we do not want to add too much flexibility, otherwise the model may overreact to stochastic underreporting. For the reporting delay smoother, we add 10 basis functions across the range of permitted values. Fewer than 10 led to the model poorly fitting the data, and more basis functions led to overfitting the data.

The resulting model, with splines used for both the epidemic curve term ($s_1(o)$) and reporting delay term ($s_2(d)$) is a simplified version of the model described in [19], with penalised cubic regression splines used instead of P-splines. However, instead of extrapolating the reporting delays outside of the observation window, we truncate using the maximum possible reporting delay, as described in Section 3.2.

### 3.4. Parametric reporting delay model

Alternatively, if the reporting delay is known to be smooth and unimodal, it may be well approximated by a parametric distribution, which are often used in describing time-delay distributions [20,21]. An advantage of this is that biases such as right-truncation or right-censoring can be explicitly corrected in the parametric survival model. In this case, we determine $f_\theta(d)$ before fitting the nowcasting model, and the parametric distribution can then be added to the model, giving

$$\log(y_d(o)) \sim \beta_0 + \mathrm{wday}(o) + \mathrm{wday}(d) + s_1(o) + \log(f_\theta(d)) + \epsilon_{od}.$$

The parametric distribution can be incorporated into the model as either a fixed effect, random effect, or an offset. We opt to use an offset, since this directly links the model to the probabilistic derivation in Section 3.2. To estimate the delay distribution and correct for the right-truncation, we apply the method from [22], which uses maximum likelihood estimation to

find the best-fitting distribution. For each case, we record two data points: the date of the event (S) and the date of report (E). Therefore, to estimate $f_\theta(\cdot)$, we need to model the likelihood of observing the data

$$P(E = e^* | S = s^*, E < T) = \frac{P(E = e^*, S = s^*)}{P(E < T, S = s^*)} = \frac{P(E = e^* | S = s^*)P(S = s^*)}{P(E < T | S = s^*)P(S = s^*)}$$

$$= \frac{P(E = e^* | S = s^*)}{P(E < T | S = s^*)} = \frac{g_\theta(e^* - s^*)}{G_\theta(T - s^*)},$$

where $g_\theta(\cdot)$ is the probability density function of the reporting delay distribution, and $G_\theta(\cdot)$ is the cumulative distribution function. We assume that the reporting delay follows a Weibull distribution. To calculate the probability of cases being reported after $d$ days, $f_\theta(d)$, we have

$$f_\theta(d) = \int_{d-1}^{d} g_\theta(\tau)d\tau = G_\theta(d+1) - G_\theta(d).$$

To fit this distribution, we use maximum likelihood estimation. We only consider the central estimate for the distribution, rather than propagating distribution uncertainty through the nowcasting model. Extensions using a Bayesian version of this method, such as [23], could be applied to correctly propagate the distribution uncertainty, for example by bootstrapping the nowcasting model across posterior samples of the reporting delay. However, such methods were not applicable since Bayesian packages could not be used on the IT systems used to access the data. Additionally, the model uncertainty is found to be well calibrated using only the central estimate, so the additional uncertainty would not substantially improve performance.

This model extends on existing GAM-based nowcasting models [19,24] through the explicit incorporation of the right-truncation corrected reporting delay distribution, $f_\theta(d)$, instead of fitting the reporting delay within the nowcasting model. This allows the model to make optimal use of individual-level data describing the reporting process.

## 3.5. Adjusting for data limitations and changes to the reporting cycle

In Section 2.3, data limitations are discussed that could bias the nowcasting. Where these limitations affect all days equivalently, the limitations can be mitigated through changing the data to reflect the true data reporting cycle, and leaving the model structure unchanged. The aim of these changes is to make the historic data reflect the current reporting practices, so that the model can detect an appropriate reporting delay distribution.

To reflect the changes around specimen processing, we amend the data prior to 30 June 2022 to have an extra day between specimen collection and reporting. For symptom onset date, we do not consider this change to the reporting cycle, since it is less sensitive to symptom onset date than it is to specimen date. However, we still need to modify the symptom onset data to reflect the difference between the date the case was reported and the date "symptom onset date" was reported. From early analysis, in the majority of cases symptom onset date was added after the case was reported. Therefore, the "reported date" will be earlier than "symptom onset date reported date"; the latter is the variable needed for nowcasting, since this reflects the delay the most recent data points are exposed to. Through this early analysis, the average reporting delay was found to be approximately two days longer than that suggested by the "reported date". Therefore, we consider a modified version of the data, where the "reported date" is changed to be two days later than the recorded "reported date".

Whilst the symptom onset date and 30 June 2022 specimen date corrections affect all days of the week equally, the data processing changes on 6 July 2022 introduced a more complicated change to the reporting cycle, changing the reporting cycle for tests processed over the

weekend. To account for this, we altered the historic data where specimen date was on Friday and Saturday to reflect this change, adding an extra day to the corresponding date of report. However, this now resulted in specimens collected on a Friday or Saturday having different reporting cycles to the other weekdays. To adjust for this in the model, in the non-parametric model, we added independent reporting delay splines to the GAM, based on whether the specimen date was a Friday or Saturday, or not. We incorporated these as fully independent smoothers, with no shared trend or smoothness [25]. This results in the model

$$\log(y_d(o)) = \beta_0 + \text{wday}(o) + \text{wday}(d) + s_1(o) + s_{2,FS(o)}(d) + \epsilon_{od},$$

where $FS(o)$ is an indicator function on whether day $o$ is either Friday or Saturday, or another day of the week. This model could be adjusted to use smoothers with either shared trends or smoothness, if appropriate. In the parametric model, we fitted $f_\theta(\cdot)$ independently depending on whether the event date was a Friday or Saturday, or other day of the week. This is conceptually similar to the non-parametric approach.

When evaluating model performance, we consider models with the amended data and models with the original data, to check whether the data driven modifications improve model accuracy. These data limitations demonstrate the importance of communicating with data collection/data processing teams when developing nowcasting methods, to ensure any large changes/complications in the data stream can be modelled appropriately.

### 3.6. Regional hierarchical model

The initial outbreak was centred in London. As the outbreak spread, cases started occurring in the other regions of England, but at much lower case numbers. The geographic range of the spread means that a single epidemic curve for the whole of England may no longer be reliable, as this could conceal regional differences in the epidemic curves. Therefore, we constructed a hierarchical model [25] to allow the epidemic curve to vary across the regions. Since the regional case counts are far lower than London, we assume that the reporting delay distribution is consistent across the country, so that all regions follow the same reporting cycle. This allows the smaller regions to borrow national information when parameterising the reporting delay, so that we do not get excessive uncertainty. We then fit independent smoothers to the epidemic curve for each region, to allow regional variation. We also incorporate a national epidemic curve to account for correlation between regions. This creates a hierarchical structure where regions have some shared trend, but different smoothness. This results in the model

$$\log(y_{d,\text{region}}(o)) = \beta_0 + \beta_{1,\text{region}} + \text{wday}(o) + \text{wday}(d) + s_1(o) + s_{2,\text{region}}(o) + s_{3,FS(o)}(d) + \epsilon_{od},$$

where $\beta_{1,\text{region}}$ is a region-specific adjustment to the intercept. For operational purposes, we opted for a two-region breakdown: London and non-London, but the method could be applied to smaller regions.

### 3.7. Confidence intervals and prediction intervals

There are two methods for quantifying uncertainty with nowcasting models: confidence intervals and prediction intervals. Confidence intervals capture the uncertainty about the expected future trends, but do not capture potential noise in the future data. Prediction intervals capture both uncertainty in the trend and noise in the data by also including uncertainty in the data generating process. That is, prediction intervals indicate the interval in which the observed future values are expected to fall. Since the mpox data are noisy, and case numbers relatively low, we focus on prediction intervals, since these provide the range of extreme data values that we might observe, which is useful for surveillance.

To calculate prediction intervals, we generate posterior samples from the fitted model, using the Metropolis-Hastings sampler provided by the gam.mh function in mgcv. For each posterior sample of the model coefficients, we calculate the expected value of the model and convert back to the linear scale. Taking these expected values, we can simulate realisations of the data generating process with these expected values. In this model, we assume that the data are sampled from a negative binomial distribution. In mgcv, the negative binomial distribution is parameterised in the terms of the mean, $\mu$, and variance, $\sigma^2$, such that $\mu = E[y]$ and $\sigma^2 = E[y] + \frac{E[y]^2}{\theta}$. In this formulation, the $\theta$ parameter is also fitted. Therefore, given the expected values of the model, $\mu$, and the fitted $\theta$ (sometimes referred to as the size parameter), we can simulate realisations of the model from a negative binomial distribution with these parameters. For each posterior sample of the coefficients, we draw a random realisation from the negative binomial model. Combining these across all posterior samples, we can calculate the prediction intervals of the model. The full posterior prediction sample is also extracted for growth rate calculations.

### 3.8. Model scoring

To assess the optimal model structures, models are scored using the scoringutils [17] package. This package produces a wide range of scoring metrics. We focus on three of these metrics when comparing the models: calibration (95% coverage)—the proportion of predictions that fall within the prediction intervals should be close to the confidence level of those intervals; bias–the probability that the predictions overestimate or underestimate the true value; interval score–a weighted measure of distance between the predicted distribution and the true observation [26].

To score the model, the predictions are compared to the observed data when sufficient time has passed to assume the data are complete. The scoring is performed over a range of lead times from 0 days to 8 days. A lead time of 0 days means that the model is trying to predict the values for the most recent date, and a lead time of 8 days means that the model is trying to predict the values for 8 days before the most recent date. For longer lead times, accuracy is expected to be higher, since the data are more complete, with the nowcasting models having to do minor adjustments. For shorter lead times, the models require very large adjustments to the observed data, so are likely to have reduced accuracy.

### 3.9. Growth rate calculation

To calculate the real-time growth rate of the epidemic, we take posterior predicted sample trajectories from the nowcasting model, and estimate the growth rate using the method described in [27,28]. That is, for each posterior nowcasting trajectory, we fit the growth rate model independently. From the growth rate model, we generate a posterior sample of growth rates using the gam.mh function in mgcv. The posterior growth rate samples are then combined across the posterior nowcasting predictions, forming a full posterior distribution for the growth rate, from which confidence intervals and central estimates are extracted.

## 4. Results

### 4.1. Operational results

**4.1.1. Nowcast and growth rates.** Throughout the mpox outbreak, these methods have been applied to monitor the status of the outbreak in real time. Figs 2 and 3 show nowcasts generated using data up to 31 August 2022 by specimen date and symptom onset date, respectively, and Fig 4 shows the growth rate by date of report. In Figs 2 and 3, the left panel shows

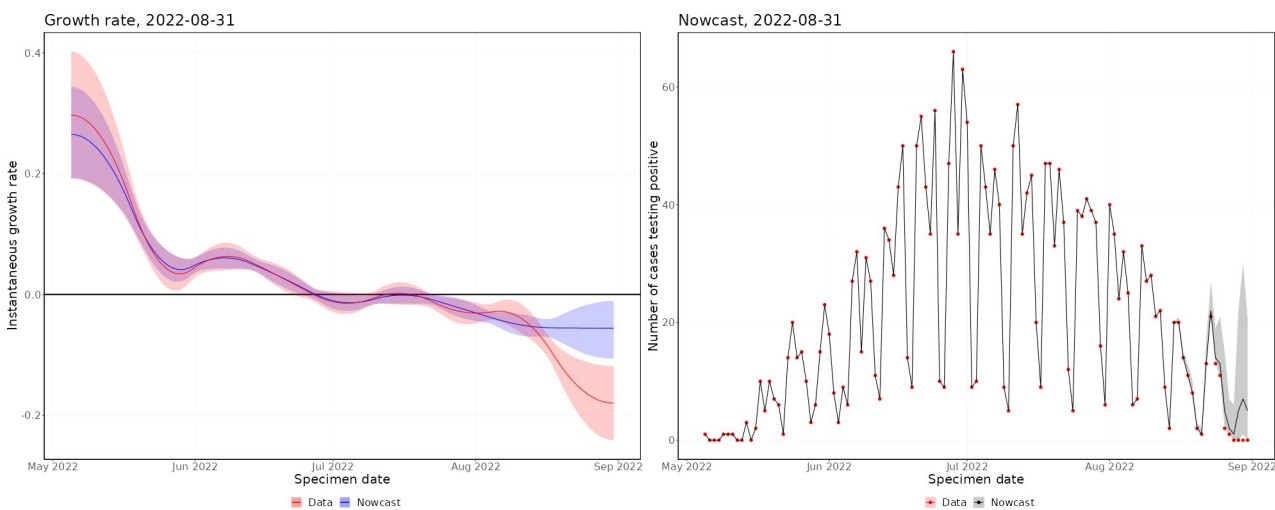

**Fig 2. Latest nowcast by specimen date.** Left panel–instantaneous growth rate, the shaded region indicates 95% confidence intervals, with the solid line indicating the median estimate. Right panel–nowcast modelled incidence, the shaded region indicates 95% prediction intervals of the model, with the solid line indicating the median nowcast.

the growth rate, calculated using the nowcast and the raw data, and the right panel shows the nowcast and the raw data. Since the delayed reporting mostly affects the most recent dates, the historic growth rates are consistent. However, across the last few weeks, the growth rates diverge significantly, with the raw data growth rate being substantially lower than the nowcast growth rate, which is expected since the raw data underestimates the recent incidence.

In May, the epidemic was growing exponentially, with an early exponential growth rate of approximately 0.3 per day (by specimen date). This was likely biased by increasing awareness leading to a burst of initial cases being reported. By the end of May, the growth rate had declined to approximately 0.05, where it remained until mid-June. After this, the growth rate started declining, until the epidemic peaked (by specimen date) at the end of June. There was quite a flat peak, with growth rates around zero from the end of June until mid-July, after

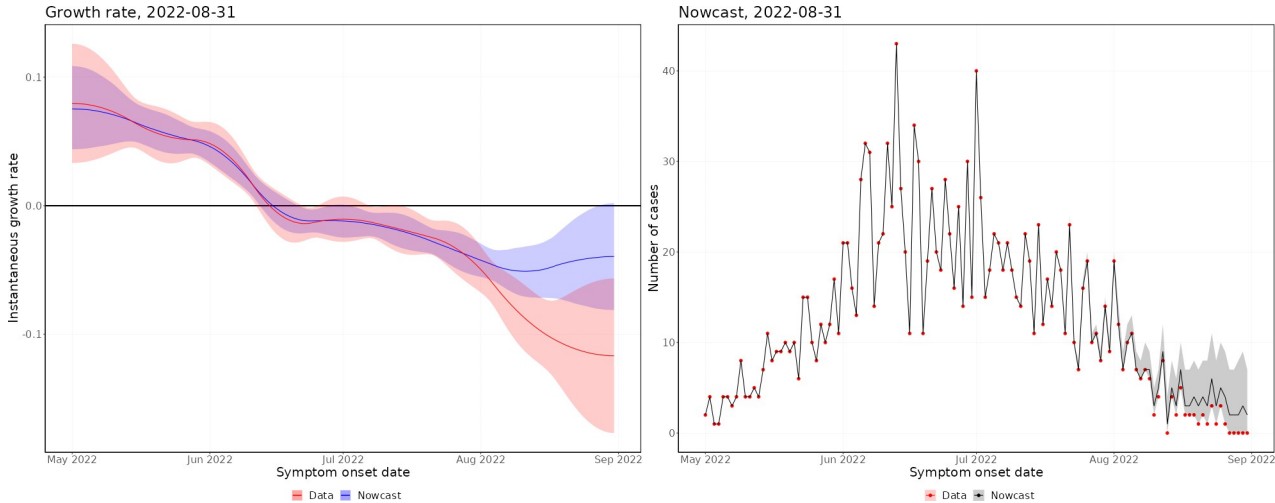

**Fig 3. Latest nowcast by symptom onset date.** Left panel–instantaneous growth rate, the shaded region indicates 95% confidence intervals, with the solid line indicating the median estimate. Right panel–nowcast modelled incidence, the shaded region indicates 95% prediction intervals of the model, with the solid line indicating the median nowcast.

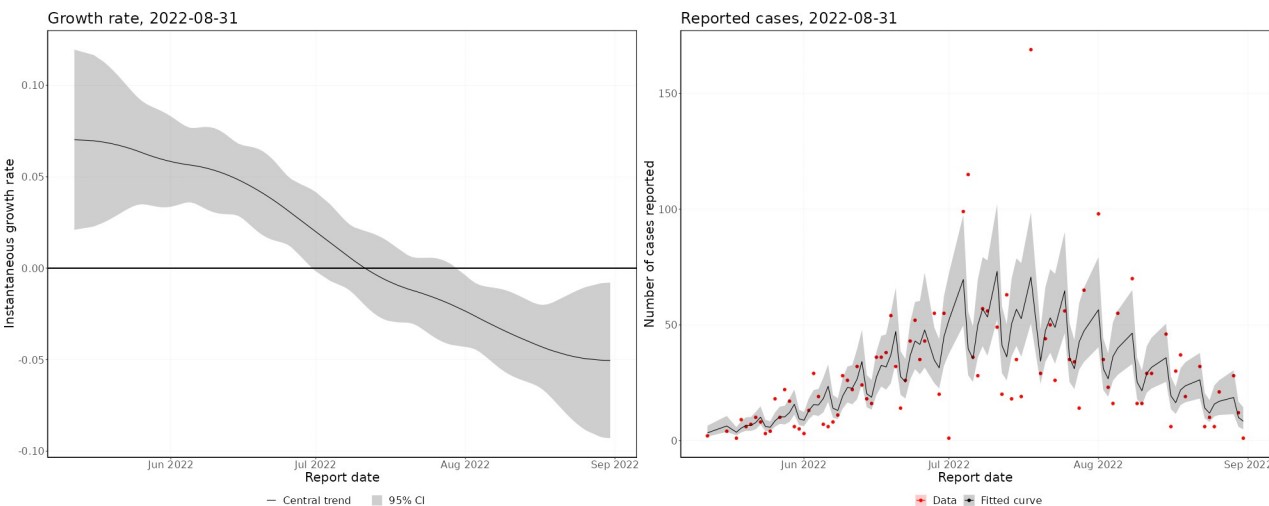

**Fig 4. Growth rate by date of report.** Left panel–instantaneous growth rate, the shaded region indicates 95% confidence intervals, with the solid line indicating the median estimate. Right panel–nowcast modelled incidence, the shaded region indicates 95% prediction intervals of the model, with the solid line indicating the median nowcast.

which the epidemic started declining. Since August, the epidemic has been relatively flat, with very low daily incidence. By symptom onset date, the growth rates follow a similar pattern, but the epidemic peaks a couple of weeks earlier. However, the peak of the symptom onset date curve is biased by the changing ascertainment rates between June and July. By date of report, the epidemic peaks later, with the peak occurring towards the start of July. This demonstrates the importance of nowcasting to understand the epidemic in real-time, with ideally the now-cast being generated from the earliest epidemiological event possible.

**4.1.2. Regional nowcast.** The mpox outbreak in the UK has seen a majority of cases being identified in London. To account for this, we constructed a regional nowcasting model (Fig 5). Here, all non-London regions have been grouped due to low incidence in these regions. The epidemic started taking off in London, with much faster initial growth rates than the other regions. However, the epidemic in London peaked earlier than the regional epidemics, where outbreaks continued after the London outbreak was declining. The peak outside London was flatter than in London, which was likely driven by different timings of peaks across the non-London regions. Regardless, most cases still occurred in London, with much smaller outbreaks in the other regions.

## 4.2. Model selection

**4.2.1 Scoring procedure.** Here we present model scoring results for a range of different models. To select the optimal model, we are interested in which model scores most favourably in 2 out of 3 of the metrics. If one of the metrics is a draw, then the magnitude of the other metrics are compared. If one metric has a substantial magnitude difference compared to the other, this metric is used to select the model. If there is no substantial magnitude difference, the calibration is taken to be the most important metric, followed by the bias.

For symptom onset date, the models are scored on data from 1 July 2022 to 31 August 2022. Data before July are excluded from the model scoring, since during June 2022 the ascertainment rate dropped rapidly, biasing the model. For specimen date, data are including from 1 June 2022 to 31 August 2022. Data prior to June 2022 are excluded as the daily counts were too low and stochastic.

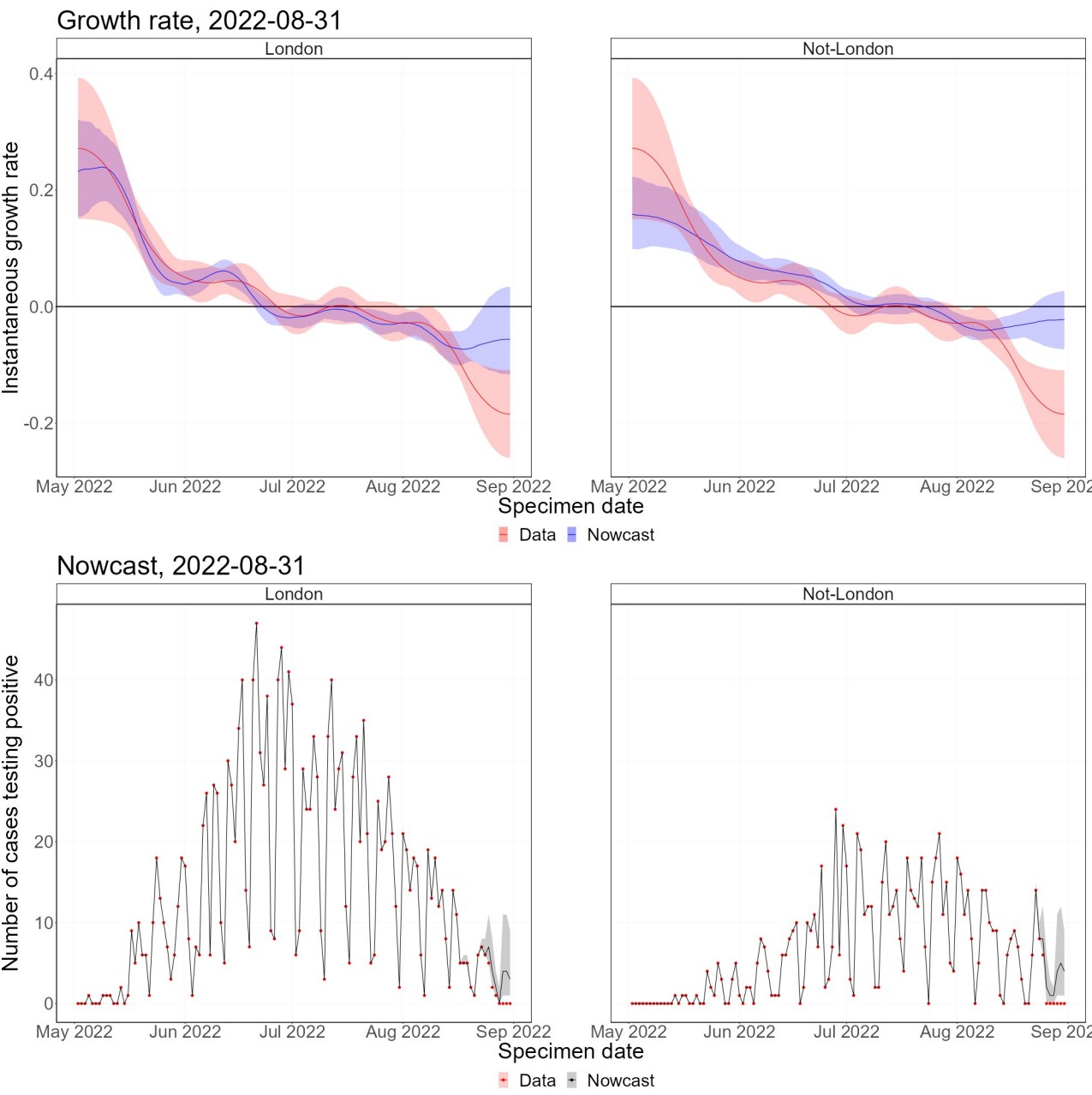

**Fig 5. Latest regional nowcast by specimen date.** Upper panels–instantaneous growth rate. Lower panels–nowcast modelled incidence. In the growth rate plots, the shaded regions indicate 95% confidence intervals, with the solid lines indicating median estimates. The shaded regions in the nowcasting plots are 95% prediction intervals of the model, with the solid line indicating the median nowcast.

To generate model scores, predictions where the predicted value was over 100 times larger than the true value (and the true value is not equal to zero) have been omitted, since these cause the interval score to grow to very large values. This only affects the non-parametric models, which appear to have some occasional numerical instability that leads to very large and unrealistic predictions. Removing these does not affect the validity of the scoring, since in practice such unrealistic predictions would be excluded manually.

**4.2.2 Scoring the input data length.** The length of data input to the model is important on two aspects. Firstly, the length of the data dictates the maximum reporting delay permitted

**Table 2. Input data length scoring.** * indicates the best performing model in each pair.

| Model | Modification | Range (days) | Days per knot | Target | Calibration (95%) | Bias | Interval score |
|---|---|---|---|---|---|---|---|
| parametric | data driven | 30 | 14 | onset date | 0.93 | -0.265 | 2.30 |
| parametric | data driven | 60* | 14 | onset date | 0.94* | 0.0352* | 2.02* |
| parametric | data driven | 30 | 14 | specimen date | 0.94 | -0.0647 | 3.96 |
| parametric | data driven | 60* | 14 | specimen date | 0.94 | -0.0604* | 2.67* |
| non-parametric | data driven | 30 | 14 | onset date | 0.93 | 0.110* | 2.41 |
| non-parametric | data driven | 60* | 14 | onset date | 0.94* | 0.187 | 2.03* |
| non-parametric | data driven | 30* | 14 | specimen date | 0.93* | 0.118* | 6.57 |
| non-parametric | data driven | 60 | 14 | specimen date | 0.98 | 0.131 | 3.98* |

to the model. Secondly, the length of the data is important for identifying trends in the epidemic curve. We consider two lengths of data, 30 days and 60 days, and perform model scoring to determine which is optimal for each model.

For both parametric models, the 60 days data range outperforms the 30 days model on all scores (Table 2). For non-parametric symptom onset date, the 60 days model outperforms the 30 days model on 2 out of 3 scores. For non-parametric specimen date, the 30 days model outperforms the 60 days model on 2 out of 3 scores.

**4.2.3. Scoring the number of knots.** The number of knots in the model controls the flexibility of the splines fitted through the event date, which controls the shape of the epidemic curve. These splines need to be flexible enough to respond to temporal change in the epidemic, but not too sensitive to overreact to noise in the reporting delays. To tune the number of knots, we define a denominator variable, which we divide the number of data points used by to obtain the number of knots. This can be interpreted as placing a knot approximately every $d$ days, where the denominator is equal to $d$. We consider two denominator values, $d = 7$ and $d = 14$. We do not consider $d > 14$ since rounding would not increase the number of knots when $T = 30$ days. For the parametric models, $d = 14$ is found to be stronger (Table 3). For non-parametric symptom onset date, both models have the same coverage, but $d = 14$ has reduced bias. For non-parametric specimen date, the $d = 7$ model outperforms $d = 14$ on all scores.

**4.2.4. Scoring the model.** Here we compare the parametric models to the non-parametric models, using the strongest configurations of data range and knots. For both target variables, the parametric models are found to be stronger (Table 4). The calibration scores (Coverage (95%)) are comparable for both models, but the interval scores are better for the parametric models and the parametric models have reduced bias.

**Table 3. Knots scoring.**

| Model | Modification | Range (days) | Days per knot | Target | Calibration (95%) | Bias | Interval score |
|---|---|---|---|---|---|---|---|
| parametric | data driven | 60 | 14* | onset date | 0.94* | 0.0352 | 2.02* |
| parametric | data driven | 60 | 7 | onset date | 0.93 | 0.0018* | 2.10 |
| parametric | data driven | 60 | 14* | specimen date | 0.94 | -0.0604* | 3.67* |
| parametric | data driven | 60 | 7 | specimen date | 0.95* | -0.0929 | 3.86 |
| non-parametric | data driven | 60 | 14* | onset date | 0.94 | 0.187* | 2.03 |
| non-parametric | data driven | 60 | 7 | onset date | 0.96 | 0.213 | 1.98* |
| non-parametric | data driven | 30 | 14 | specimen date | 0.93 | 0.118 | 6.57 |
| non-parametric | data driven | 30 | 7* | specimen date | 0.94* | 0.102* | 5.97* |

* indicates the best performing model in each pair.

**Table 4. Parametric versus non-parametric scoring.**

| Model | Modification | Range (days) | Days per knot | Target | Calibration (95%) | Bias | Interval score |
|---|---|---|---|---|---|---|---|
| Non-parametric | data driven | 30 | 14 | onset date | 0.94 | 0.187 | 2.03 |
| parametric* | data driven | 60 | 14 | onset date | 0.94 | 0.0352* | 2.02* |
| Non-parametric | data driven | 30 | 7 | specimen date | 0.94 | 0.102 | 5.97 |
| parametric* | data driven | 60 | 14 | specimen date | 0.94 | -0.0604* | 3.67* |

* indicates the best performing model in each pair.

**4.2.5. Scoring the data modification.** Through discussions with data teams, we were able to understand temporal changes in reporting patterns. Additionally, we were able to measure extra reporting delays that were not identifiable from the data, due to the lack of time stamped database extracts. This led to two variations of the data considered in the model–one where the data have been modified to incorporate the reporting structure (adding the delay from case-reporting to symptom-onset-reporting for symptom onset date, and adding the change to the specimen processing for specimen date) and one with the raw data. Using the best scoring model from Section 4.2, we score the different data sets in Table 5. The modified data is favourable for both symptom onset date and specimen date. For onset date, both models have comparable coverage, but the modified data leads to reduced bias. For specimen date, the modified data leads to an improvement on all three scores.

By symptom onset date, there is consistently improved performance in the modified data, since on average the reporting delay suggested by the raw data is 2 days shorter than the true reporting delay, which is corrected in the modified data. By specimen date, the reporting patterns changed around the end of June. Therefore, prior to this, and sufficiently after this, the two data sets are consistent. However, around early July, the modified data substantially outperforms the raw data, which is demonstrated in the overall improved performance.

## 4.3. Model performance

**4.3.1. Historical trajectories.** Early in the outbreak, the day of week effects in the specimen date curve were unstable. This is reflected in Fig 6, which shows the nowcast generated on 2 June 2022, relative to the complete data for those dates. The nowcast captures the general upwards trend in the curve, but struggles to capture the magnitude of the day of week effects of the recent data points. This is because the data has only entered a stable day of week cycle over the last week, and the model is attempting to fit day of week effects to the last 60 days, so is biased by the early trends. As the outbreak progressed, the day of week cycle became more stable, and the model was able to capture this, as shown in the nowcast generated on 25 July 2022 (Fig 6). Here, the nowcast is able to recreate the day of week cycle in the recent data. The model also captures the general trend of the outbreak, with all values within the prediction intervals and close to the central estimate.

**Table 5. Data modification scoring.**

| Model | Modification | Range (days) | Days per knot | Target | Calibration (95%) | Bias | Interval score |
|---|---|---|---|---|---|---|---|
| parametric | none | 60 | 14 | onset date | 0.96 | 0.141 | 1.41* |
| parametric | data driven* | 60 | 14 | onset date | 0.94 | 0.0352* | 2.02 |
| parametric | none | 60 | 14 | specimen date | 0.91 | -0.136 | 3.77 |
| parametric | data driven* | 60 | 14 | specimen date | 0.94* | -0.0604* | 3.67* |

* indicates the best performing model in each pair.

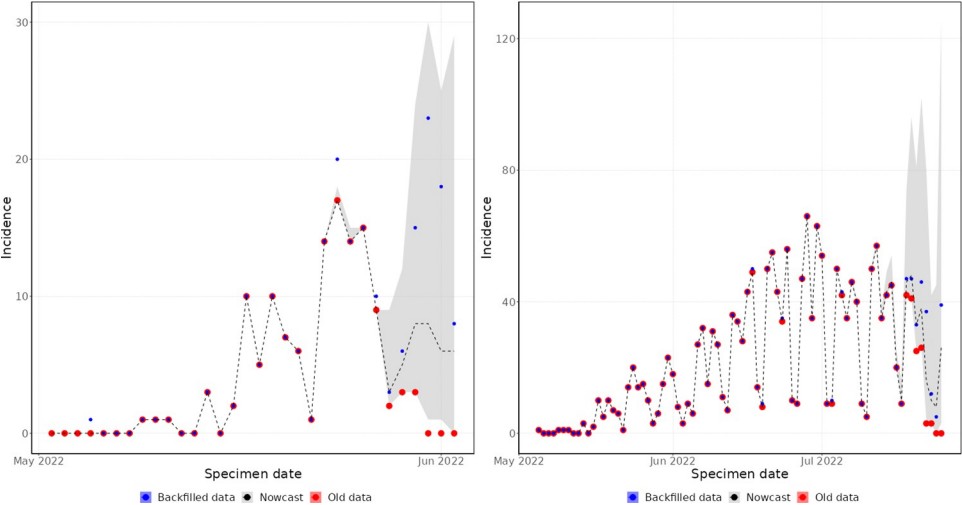

**Fig 6. Specimen date historic nowcasting.** Left panel–nowcast generated on 2 June 2022. Right panel–nowcast generated on 25 July 2022. Old data (red) are those available at the time the nowcast was produced. Backfilled data (blue) is the true value of the data. The shaded region (grey) is the 95% prediction interval of the nowcasting model, with median value marked by the dashed line.

By symptom onset date, the day of week effects do not have such a strong influence on the data. Therefore, early in the outbreak, the model performs well at predicting the shape of the epidemic, as reflected in the nowcast generated on 2 June 2022 (Fig 7). Here, the nowcast captures the rising trend of the epidemic, but with very high uncertainty due to the small number of historic observations and the long delay from symptom onset date to reporting date. The nowcast generated on 25 July 2022 was able to predict the data more accurately (Fig 7), since there were more historic data from which to accurately estimate the reporting delay and epidemic trend.

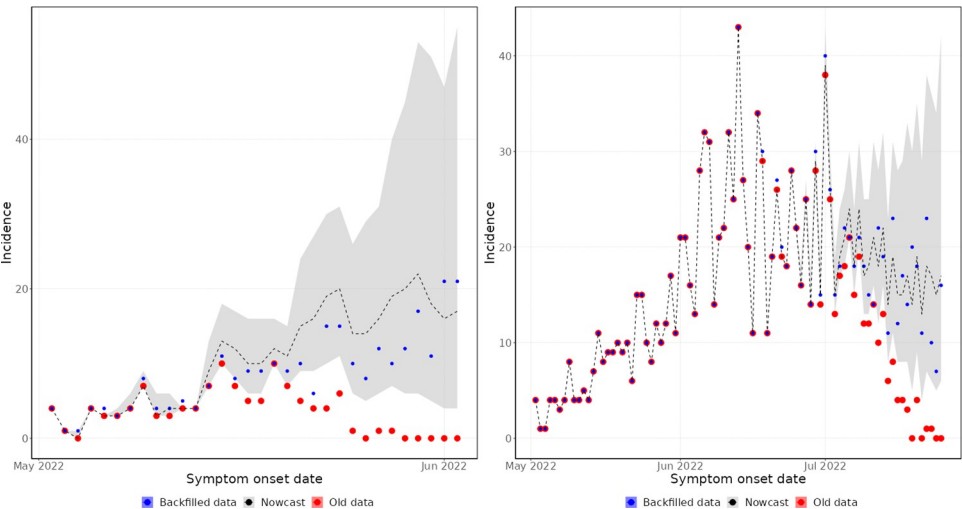

**Fig 7. Symptom onset date historic nowcasting.** Left panel–nowcast generated on 2 June 2022. Right panel–nowcast generated on 25 July 2022. Old data (red) are those available at the time the nowcast was produced. Backfilled data (blue) is the true value of the data. The shaded region (grey) is the 95% prediction interval of the nowcasting model, with median value marked by the dashed line.

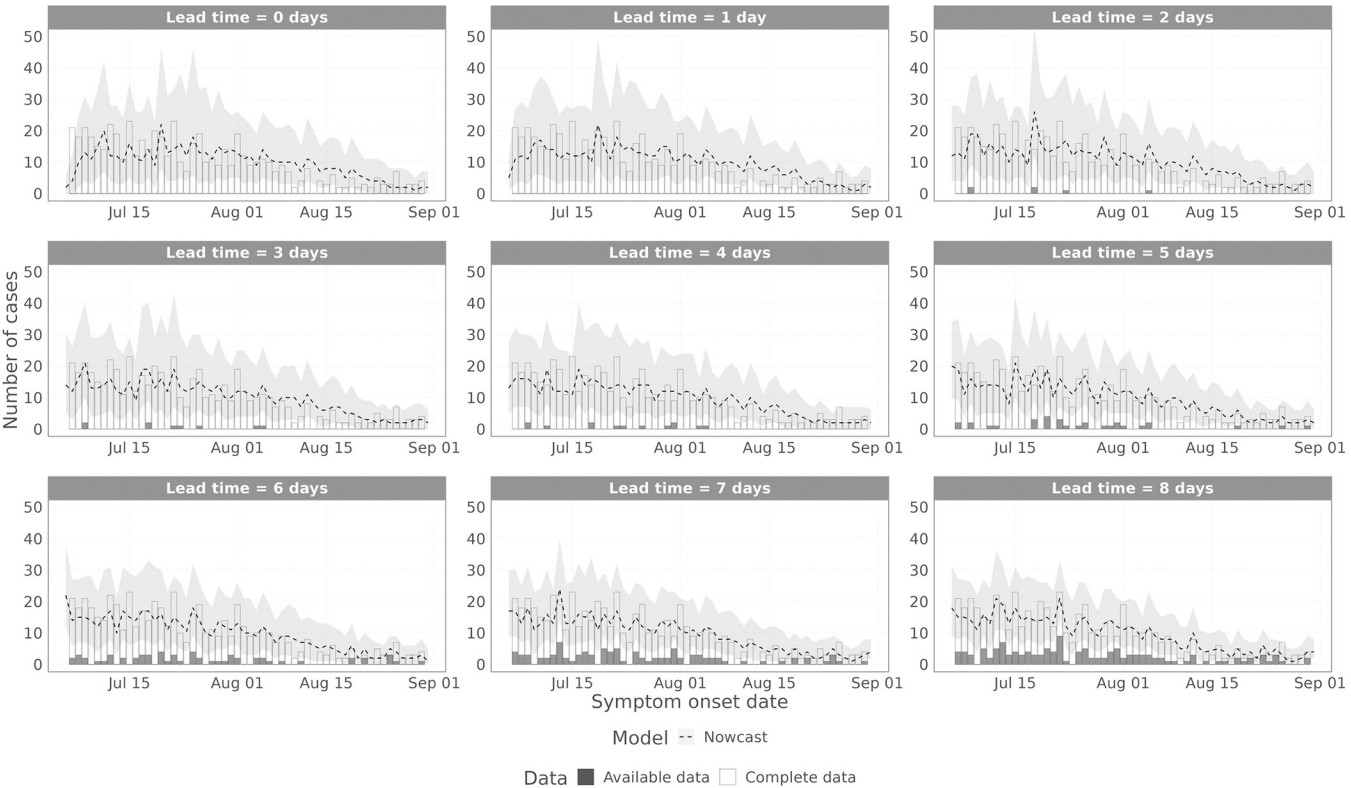

**Fig 8. Performance of the non-parametric and parametric nowcasting models by symptom onset date.** The grey bars indicate the data available at the time of the nowcast, the white bars are the complete data, and the lines are the nowcasting projection, with 95% prediction intervals in the ribbon. Each panel shows a different lead time, where the nowcasting model is trying to predict that many days prior to the current date.

**4.3.2. Symptom onset date performance.** To visualise the model performance over time, we plot the model estimated values to the complete data and the data that were available at the time (Figs 8 and 9 for symptom onset date and specimen date, respectively), for different lead time values, i.e. the difference between the date being estimated and the date the nowcast was generated. Fig 8 shows the results for the symptom onset date model scoring periods (1 July 2022–31 August 2022). S1 Fig shows the symptom onset date results including June 2022. For symptom onset date, there are substantial reporting delays, so with lead times up to 8 days, the majority of data points remain incomplete. Therefore, for all these lead times considered, the nowcasting model has to do substantial inference. From mid-July onwards (Fig 8), the model captures the trend in the data at all lead times, with performance increasing at longer lead times. However, from mid-June to mid-July (S1 Fig), the model struggles, first overestimating the data before underestimating the data. This is expected, since during this period the ascertainment rate of data on symptom onset date declined rapidly. Therefore, whilst the epidemic was still likely growing in mid-June, the data started to slow down and decline, accelerated by the declining ascertainment rate. Eventually the model captured this decline, but in early July the ascertainment rate stabilised, so the data stopped declining, whereas the model continued the observed trend in the data, leading to underestimation.

Looking at the bias in predicting the number of cases each day (Fig 10 –lower panel), there is a phase of biased predictions during mid-August 2022, with the model persistently overestimating the true number of cases (Fig 8). This appears to be caused by a sudden drop in the growth rate during August 2022 (Fig 3), which the model is slow to capture.

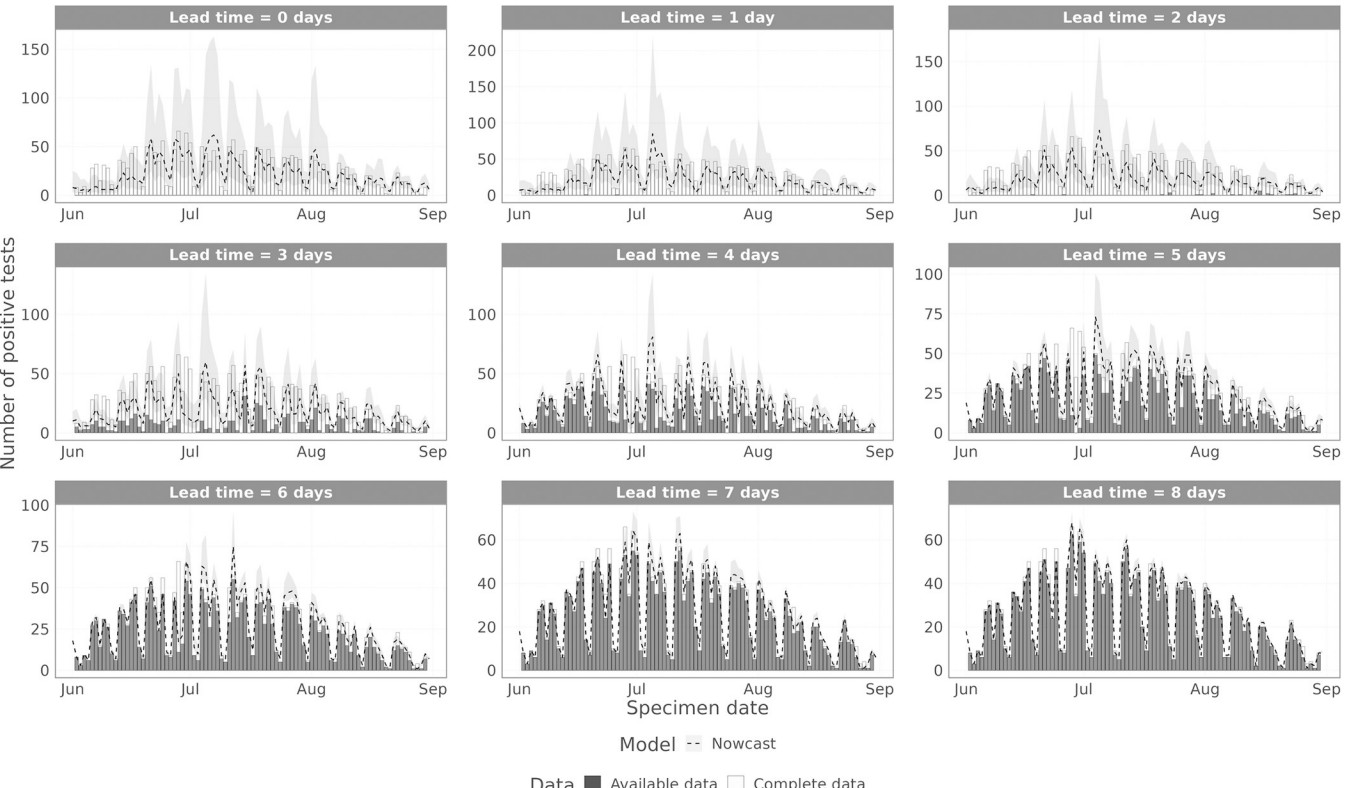

**Fig 9. Performance of the non-parametric and parametric nowcasting models by specimen date.** The grey bars indicate the data available at the time of the nowcast, the white bars are the complete data, and the lines are the nowcasting projection, with 95% prediction intervals in the ribbon. Each panel shows a different lead time, where the nowcasting model is trying to predict that many days prior to the current date.

**4.3.3. Specimen date performance.** When nowcasting by specimen date, the reporting delays are much shorter than when using symptom onset date. This can be seen in Fig 9, where the data are almost complete at lead times of 7 and 8 days, with data completeness at a lead time of 3 days comparable to symptom onset date with a lead time of 8 days. The model captures the trend in the epidemic curve for all lead times. After mid-June, the model also reproduces the day of week effect in the data, but prior to this the model does not, as there were insufficient data to estimate the day of week effect. For longer lead times, the performance improves, and the prediction intervals become tighter. The bias when predicting the number of positive cases by specimen date (Fig 10) shows no consistent bias over time, with the model performing comparably across the whole time period.

## 5. Discussion

The outbreak in 2022 has demonstrated for the first time the potential for mpox to lead to major outbreaks outside of Central and Western Africa. At a time where the COVID-19 pandemic is ongoing and has placed substantial pressure on healthcare systems worldwide [29], timely and accurate surveillance was essential for understanding the pressures of mpox on these systems. Timely surveillance relies on how temporally close our surveillance data reflects the epidemic curve, so that interventions and changes in the epidemic can be detected and evaluated.

In the UK, the three main surveillance dates of interest are symptom onset, specimen, and reporting. Reporting date is the most lagged relative to the epidemic curve, and symptom

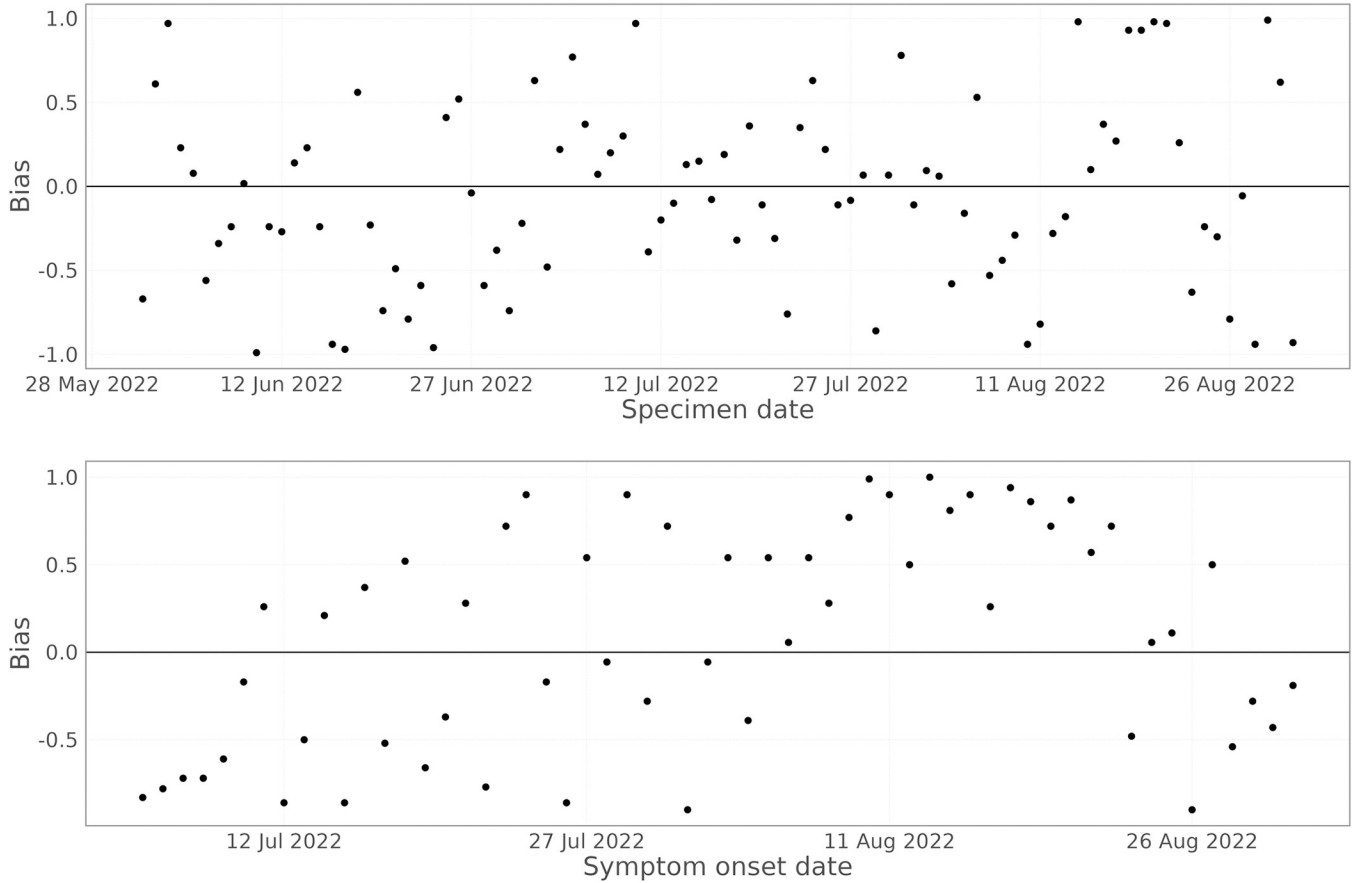

**Fig 10. Bias in the nowcasting model over time, averaged over lead times from 0 days to 8 days.** The upper panel shows the results for specimen date nowcasting, the lower panel the results for symptom onset date nowcasting.

onset date is the least [30]. However, the coverage of data on symptom onset date has fluctuated throughout the epidemic, declining from very high (over 90%) coverage before June 2022 to 50% coverage in August 2022. This can lead to a biased epidemic curve if not adjusted for appropriately. Although more delayed, specimen date can be a valuable surveillance data stream as it is less prone to ascertainment bias and less dependent on an individual's detection and recall of symptom onset.

Since there is a delay from symptom onset and specimen date to reporting date, these data streams are subject to backfilling, where occurrences on recent dates will not be detected until those occurrences are reported. Therefore, nowcasting methods, which aim to predict the missing occurrences, are essential. There are several nowcasting methods in the epidemic modelling literature that are well principled and implemented in Bayesian frameworks [6,7]. However, this leads to them lacking flexibility to modify easily. In many nowcasting problems, the unique characteristics of the local setting and data mean that a general tool may not be the best approach. For example, through having access to individual level data on mpox patients, we were able to robustly estimate the reporting delay distributions using parametric distributions. Being able to include these parametric distributions directly in the nowcasting model substantially improved performance and stability of the model. Additionally, computational tractability and reliability were vital in ensuring continued delivery of the nowcasting products each week. The Bayesian implementation of existing leading models build on popular Bayesian

packages such as stan and jags. However, the multiple levels of permissions required created issues on the managed IT systems used within UKHSA to handle the highly sensitive mpox data meant that these could not be used, which would have prevented efficient modelling pipelines to be built, which is essential for stable product delivery.

Therefore, we opted to create a novel nowcasting framework using generalised additive models, with sufficient flexibility to capture the evolving data environments and to make maximal use of the individual-level data available. The resulting models are relatively simple, given functionality to implement complex splines in a robust fashion, compared to the Bayesian epidemic models, but it is easy to implement and efficient to run. These models demonstrate robust performance, with the predicted values generally closely mapping the shape of the epidemic, and the estimated growth rates vastly improving on those estimated from the raw data. The penalised splines used effectively result in the model predicting a continued exponential trend in the data, unless otherwise informed by the partially complete data across the most recent data points (once the reporting delay is accounted for). The proposed models build upon similar GAM-based nowcasting models that have been used for nowcasting hospital admissions with COVID-19 in Germany [13] through the explicit incorporation of the right-truncation corrected reporting delay distribution, $f_\theta(d)$, which allows the model to make optimal use of individual-level data describing the reporting process.

The proposed models were used weekly throughout the mpox outbreak in England, facilitating reliable estimation of timely growth rate estimates, to understand how the outbreak is evolving [14]. From the analysis, we can see the clear difference in the epidemic curve across the different data streams. Data on symptom onset date shows the epidemic curve peaking earliest. However, the exact timing of the peak based on symptom onset date is unreliable, since the ascertainment rate of this data was declining at the same time as the peak. Specimen date on the other hand has been reliably recorded throughout the epidemic, leading to a reliably timed peak at the end of June. Similarly, date of report is consistently recorded, allowing reliable timing of the peak, but is further lagged, occurring 1–2 weeks later than the specimen date peak. This demonstrates the value of nowcasting methods, since they can detect the epidemic peak earlier.

There are a variety of complications arising from the data that have made the nowcasting process more challenging. The key complications are changing ascertainment of data on symptom onset date and changing data processing procedures. The changing ascertainment around symptom onset leads to a period where the nowcasting model struggles to perform, since the case data do not reflect the underlying trend of the epidemic. This is echoed in S1 Fig, where performance can be seen to drop during the period of changing ascertainment rates. Future research into nowcasting methods that can also adjust for changing ascertainment rates is a valuable direction for the field. Changes to data processing procedures led to a scenario where at certain points in time, the relationship between an event occurring and being reported can change. To adjust for this, we have attempted to modify the historic data to reflect the new processing structure. This modification was based on discussion with the data collection and processing teams, to ascertain how the new structure will vary. Such an approach appears to work here, since no noticeable drop in performance is observed around the change point. This illustrates the importance of communicating with data collection teams when developing nowcasting methods. Further research into developing methods to accurately correct this, either through data modification or coding the discrepancies into the model, are a vital future direction in nowcasting research.

## 6. Conclusion

Nowcasting in epidemic models has been studied for many years, with a range of packages available to tackle general nowcasting problems. Unfortunately, many high-quality existing

tools were not applicable in our setting, due to a lack of generalisability. Therefore, a novel method was developed, the strength of which came from modellers being embedded with the data collection of processing teams, which allowed changes to data processing to be built into the model in real-time and the model to be tailored to the nature of the data processing. The ease of implementation of this method makes this a valuable addition to the nowcasting literature. Through real-time modelling of the mpox outbreak, we identified many challenges. Future research within nowcasting focusing on methods to correct for these real-time challenges, as well as greater portability of existing nowcasting tools, is a valuable future direction within the field.

## Supporting information

**S1 Fig. Performance of the non-parametric and parametric nowcasting models by symptom onset date–full study period.** The grey bars indicate the data available at the time of the nowcast, the white bars are the complete data, and the lines are the nowcasting projection, with 95% prediction intervals in the ribbon. Each panel shows a different lead time, where the nowcasting model is trying to predict that many days prior to the current date. (PDF)

## Acknowledgments

The authors would like to thank colleagues in the UKHSA mpox modelling group, UKHSA mpox technical group, and JUNIPER consortium for various discussions around surveillance and nowcasting.

## Author Contributions

**Conceptualization:** Christopher E. Overton, Sam Abbott, Fergus Cumming, Thomas Ward.

**Data curation:** Rachel Christie, Julie Day, Owen Jones, Charlie Turner.

**Formal analysis:** Christopher E. Overton.

**Methodology:** Christopher E. Overton, Sam Abbott, Rob Paton.

**Software:** Christopher E. Overton.

**Visualization:** Christopher E. Overton.

**Writing – original draft:** Christopher E. Overton, Sam Abbott, Rachel Christie, Julie Day, Owen Jones, Rob Paton, Thomas Ward.

**Writing – review & editing:** Christopher E. Overton, Sam Abbott, Fergus Cumming, Thomas Ward.

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
