## [Decision Letter · Decision Letter 0]

12 May 2023

Dear Dr Overton,

Thank you very much for submitting your manuscript "Nowcasting the 2022 mpox outbreak in England" for consideration at PLOS Computational Biology. As with all papers reviewed by the journal, your manuscript was reviewed by members of the editorial board and by several independent reviewers. The reviewers appreciated the attention to an important topic. Based on the reviews, we are likely to accept this manuscript for publication, providing that you modify the manuscript according to the review recommendations.

In particular, the reviewers suggest that you provide additional information about the motivation and unique contribution(s) of the study to the nowcasting literature and discuss how your approach compares to other nowcasting efforts. They also suggest a number of references for your consideration; however, note that adding these citations is not required.

Sincerely,

Eric HY Lau, Ph.D.

Academic Editor

PLOS Computational Biology

Virginia Pitzer

Section Editor

PLOS Computational Biology

Reviewer's Responses to Questions

**Comments to the Authors:**

Reviewer #1: The review is uploaded as an attachment.

Reviewer #2: The article investigates a Nowcasting the 2022 mpox outbreak in England

Please add the main findings and objective of the current study in the abstract.

- What are the benchmark cases in your Nowcasting the 2022 mpox outbreak in Englandstudy?

- What are the special cases of your study?

- Table needs to be referenced.

- Main equations and propositions need to be referenced.

- Punctuation is missing after some equations.

- Try to show more of the physical situation in the results and discussion to reflect the proposed notion of the wholestudy.

- For enhancing the introduction section with the new publications, old references may be replaced with new ones such as:

https://doi.org/10.1140/epjp/s13360-023-03865-x

https://doi.org/10.1007/s40808-022-01607-z

https://doi.org/10.1088/1402-4896/ac7ebc

https://doi.org/10.1007/s40808-021-01313-2

Reviewer #3: Title: Nowcasting the 2022 mpox outbreak in England

Author: Christopher E. Overton et al.

Recommendation:

In the submitted manuscript, the authors studied a Nowcasting the 2022 mpox outbreak in England. In my opinion, the results presented in this paper are interesting and I suggest that the paper will be accepted for publication in the journal if the authors can make some revisions according to the following comments.

1- Authors can add the following references to enrich the introductory section: Mathematical analysis of spread and control of the novel corona virus (COVID-19) in China." Chaos, Solitons and Fractals 141 (2020): 110286.

On Analysis of fractional order mathematical model of Hepatitis B using Atangana–Baleanu Caputo (ABC) derivative." Fractals (2021): 2240017.

Caputo type fractional operator applied to Hepatitis B system." Fractals (2021):

The stochastic bifurcation analysis and stochastic delayed optimal control for epidemic model with general incidence function." Chaos: An Interdisciplinary Journal of Nonlinear Science - AIP (2021): 104649.

2- The English writing of the paper is required to be improved. Please check the manuscript carefully for typos and grammatical errors. Also, the English structure of the article, including punctuation, semicolon, and other structures, must be carefully reviewed.

3. They should double check the mathematical formulations, and also add appropriate references for governing equations.

**Have the authors made all data and (if applicable) computational code underlying the findings in their manuscript fully available?**

Reviewer #1: Yes

Reviewer #2: None

Reviewer #3: Yes

PLOS authors have the option to publish the peer review history of their article (what does this mean?). If published, this will include your full peer review and any attached files.

Reviewer #1: No

Reviewer #2: No

Reviewer #3: No

Figure Files:

Data Requirements:

Reproducibility:

References:

---

## [Decision Letter · Decision Letter 1]

7 Aug 2023

Dear Dr Overton,

Thank you very much for submitting your manuscript "Nowcasting the 2022 mpox outbreak in England" for consideration at PLOS Computational Biology. As with all papers reviewed by the journal, your manuscript was reviewed by members of the editorial board and by several independent reviewers. The reviewers appreciated the attention to an important topic. Based on the reviews, we are likely to accept this manuscript for publication, providing that you modify the manuscript according to the review recommendations.

The Authors have addressed all the criticisms by all Reviewers. However I have some remaining comments:

1. In Tables 2-4, could the authors explain why interval score for the non-parametric model can be that high (in the order of 1E17)?

2. Could that be dominated by one or two extreme values which may have dominated the assessment?

Sincerely,

Eric HY Lau, Ph.D.

Academic Editor

PLOS Computational Biology

Virginia Pitzer

Section Editor

PLOS Computational Biology

The Authors have addressed all the criticisms by all Reviewers. However I have some remaining comments:

1. In Tables 2-4, could the authors explain why interval score for the non-parametric model can be that high (in the order of 1E17)?

2. Could that be dominated by one or two extreme values which may have dominated the assessment?

Reviewer's Responses to Questions

**Comments to the Authors:**

Reviewer #1: All of my comments have been thoroughly addressed. Thank you!

**Have the authors made all data and (if applicable) computational code underlying the findings in their manuscript fully available?**

Reviewer #1: Yes

PLOS authors have the option to publish the peer review history of their article (what does this mean?). If published, this will include your full peer review and any attached files.

Reviewer #1: **Yes: **Fanny Bergström

Figure Files:

Data Requirements:

Reproducibility:

References:

---

## [Editor Report · Decision Letter 2]

25 Aug 2023

Dear Dr Overton,

We are pleased to inform you that your manuscript 'Nowcasting the 2022 mpox outbreak in England' has been provisionally accepted for publication in PLOS Computational Biology.

Best regards,

Eric HY Lau, Ph.D.

Academic Editor

PLOS Computational Biology

Virginia Pitzer

Section Editor

PLOS Computational Biology

Thanks for addressing all the reviewers' comments. Congratulations on the excellent work!

---

## [Editor Report · Acceptance letter]

15 Sep 2023

PCOMPBIOL-D-23-00266R2 

Nowcasting the 2022 mpox outbreak in England

Dear Dr Overton,

I am pleased to inform you that your manuscript has been formally accepted for publication in PLOS Computational Biology. Your manuscript is now with our production department and you will be notified of the publication date in due course.

With kind regards,

Jazmin Toth
